# Metabolomic Predictors of Dysglycemia in Two U.S. Youth Cohorts

**DOI:** 10.3390/metabo12050404

**Published:** 2022-04-29

**Authors:** Wei Perng, Marie-France Hivert, Gregory Michelotti, Emily Oken, Dana Dabelea

**Affiliations:** 1Lifcourse Epidemiology of Adiposity and Diabetes (LEAD) Center, University of Colorado Anschutz Medical Campus, Aurora, CO 80045, USA; 2Department of Epidemiology, Colorado School of Public Health, University of Colorado Anschutz Medical Campus, Aurora, CO 80045, USA; dana.dabelea@cuanschutz.edu; 3Division of Chronic Disease Research Across the Lifecourse (CoRAL), Department of Population Medicine, Harvard Pilgrim Health Care Institute, Harvard Medical School, Boston, MA 02215, USA; mhivert@partners.org (M.-F.H.); emily_oken@harvardpilgrim.org (E.O.); 4Diabetes Unit, Massachusetts General Hospital, Boston, MA 02114, USA; 5Metabolon, Inc., Chapel Hill, NC 27599, USA; gmichelotti@metabolon.com; 6Department of Nutrition, T. H. Chan Harvard School of Public Health, Boston, MA 02115, USA; 7Department of Pediatrics, University of Colorado School of Medicine, Aurora, CO 80045, USA

**Keywords:** metabolomics, dysglycemia, impaired fasting glucose, youth type 2 diabetes, predictive model

## Abstract

Here, we seek to identify metabolite predictors of dysglycemia in youth. In the discovery analysis among 391 youth in the Exploring Perinatal Outcomes among CHildren (EPOCH) cohort, we used reduced rank regression (RRR) to identify sex-specific metabolite predictors of impaired fasting glucose (IFG) and elevated fasting glucose (EFG: Q4 vs. Q1 fasting glucose) 6 years later and compared the predictive capacity of four models: Model 1: ethnicity, parental diabetes, in utero exposure to diabetes, and body mass index (BMI); Model 2: Model 1 covariates + baseline waist circumference, insulin, lipids, and Tanner stage; Model 3: Model 2 + baseline fasting glucose; Model 4: Model 3 + baseline metabolite concentrations. RRR identified 19 metabolite predictors of fasting glucose in boys and 14 metabolite predictors in girls. Most compounds were on lipid, amino acid, and carbohydrate metabolism pathways. In boys, no improvement in aurea under the receiver operating characteristics curve AUC occurred until the inclusion of metabolites in Model 4, which increased the AUC for prediction of IFG (7.1%) from 0.81 to 0.97 (*p* = 0.002). In girls, %IFG was too low for regression analysis (3.1%), but we found similar results for EFG. We replicated the results among 265 youth in the Project Viva cohort, focusing on EFG due to low %IFG, suggesting that the metabolite profiles identified herein have the potential to improve the prediction of glycemia in youth.

## 1. Introduction

Youth-onset type 2 diabetes (T2D) is on the rise worldwide, accounting for one in three cases of incident diabetes [1]. Common clinical assessment of T2D risk in youth relies on body mass index (BMI), waist circumference, race/ethnicity, family history of diabetes, and for high-risk youth, fasting glucose levels. However, these assessments may have limited preventive utility due to low sensitivity (e.g., BMI [2]) and because studies in adults have shown that appreciable β-cell dysfunction has already occurred by the time glycemia becomes impaired [3].

Metabolomics is a powerful tool for identifying functional biomarkers of chronic disease risk that not only aid in prediction but also shed light on pathophysiology [4]. In 2011, Wang et al. [5] identified a circulating branched chain amino acid (BCAA) “metabolomic signature”, indicative of perturbed lipid and amino acid metabolism [6], that was detectable 12 years prior to the development of insulin resistance and incident T2D among BMI-matched adults. Consequently, Wurtz et al. [7] showed that elevations in BCAA preceded worsening insulin resistance in young adults, confirming that specific metabolite profiles that reflect early pathogenic processes manifest before conventional T2D biomarkers and, therefore, may serve as early flags for disease risk.

In youth, most papers on metabolite profiles of metabolic disease risk are cross-sectional [8,9,10,11,12], making it difficult to infer whether the metabolite profiles precede or occur concurrently with conventional T2D biomarkers. Some prospective investigations provide support for the temporal sequence of events, including: an analysis of 17 adolescents that reported that elevated BCAA preceded the development of insulin resistance over 18 months [13]; a study in Project Viva that showed associations of two metabolite patterns (BCAA and androgen hormones) with subsequent change in fasting glucose, triglycerides, and adipokines from age 7 to 13 years [14]; and an analysis in the Early Life Exposure in Mexico to ENvironental Toxicants (ELEMENT) [15] Project showing that BCAA at age 10 years predicted metabolic syndrome components five years later [16]. However, to our knowledge, no study to date has formally evaluated the utility of metabolites to improve the prediction of glycemia outcomes in youth.

Here, we seek to identify sex-specific metabolomic biomarkers of dysglycemia over six years of follow-up among general-risk youth in Colorado (the EPOCH study); we also evaluate the extent to which the metabolomic biomarkers improve the prediction of dysglycemia beyond known risk factors (i.e., race/ethnicity, family history of T2D, in utero exposure to gestational diabetes, body mass index) and conventional biomarkers of T2D risk used in research settings (i.e., waist circumference, fasting insulin, lipid profile, pubertal status), and baseline glycemia (fasting glucose). Secondarily, we seek to assess the consistency of associations in an independent cohort of youth in Massachusetts (Project Viva).

## 2. Result

### 2.1. Background Characteristics and Descriptive Statistics for Both Cohorts

EPOCH participants (discovery) were aged 10.1 ± 1.4 years at baseline and 16.3 ± 1.2 years at follow-up. Approximately half (49.6%) of the sample was female and 35.5% identified as Hispanic. At baseline, the average body mass index (BMI) z-score was 0.16 ± 1.15 for girls and 0.29 ± 1.26 for boys. Baseline fasting glucose levels were 4.5 ± 0.6 mmoL/L for girls and 4.6 ± 0.9 mmoL/L for boys. At follow-up, fasting glucose was 5.0 ± 1.7 mmoL/L for girls and 5.1 ± 1.8 for boys. Prevalence of IFG was 3.1% in girls and 7.1% in boys; IGT was 3.3% in girls and 2.2% in boys; and combined dysglycemia was 4.6% in girls and 9.1% in boys. Fifteen percent of girls and 15.7% of boys had a family history of T2D.

Project Viva participants were 12.9 ± 0.6 years at baseline and 17.6 ± 0.6 years at follow-up (Appendix A). Approximately half (47.6%; *n* = 126) the sample was female; and 5.3% (*n* = 14) identified as Hispanic. At baseline, the BMI z-score was 0.52 ± 1.25 for girls and 0.81 ± 1.25 for boys. Baseline fasting glucose levels were 4.9 ± 0.6 mmoL/L for girls and 5.0 ± 0.5 mmoL/L for boys. At follow-up, fasting glucose levels were 4.6 ± 0.3 mmoL/L for girls and 4.8 ± 0.4 mmol/L for boys; 0.8% (*n* = 1) of girls and 3.6% (*n* = 5) of boys classified as IFG. No boys had a family history of T2D, but 1.6% (*n* = 2) of girls did; 4% (*n* = 5) of girls and 2.9% (*n* = 4) of boys were exposed to gestational diabetes mellitus (GDM).

### 2.2. Results from EPOCH

Table 1 shows the mean ± SD of fasting glucose at follow-up across categories of sociodemographic characteristics and conventional risk factors at baseline. Boys had higher fasting glucose than girls. We observed the positive associations of baseline BMI z-score, waist circumference, fasting glucose, and fasting insulin with fasting glucose at follow-up. High density lipoprotein (HDL) cholesterol was inversely related to fasting glucose at follow-up.

Using sex-specific reduced rank regression (RRR) models, we identified 19 metabolites in boys and 14 metabolites in girls measured at baseline as predictors of ln-fasting glucose at follow-up. The metabolites were on lipid, amino acid, nucleotide, and carbohydrate metabolism pathways. Table 2 displays metabolite identity and average RRR regression coefficients. Appendix A is a list of metabolites that were in the top 10% of RRR regressions coefficients during at least one of the five iterations.

Appendix A show associations of each metabolite at baseline with ln-fasting glucose at follow-up. We noted weak-to-moderate β-estimates that were consistent across models in both sexes. The weak associations reflect the fact that RRR identifies a set of predictors that together (rather than individually) optimize variability in the outcome.

Table 3 shows the performance of the four models in predicting impaired fasting glucose (IFG), elevated fasting glucose (EFG, defined as Q4 vs. Q1 of fasting glucose at follow-up), and combined dysglycemia (defined as IFG, impaired glucose tolerance, or T2D—see Methods section for details). In boys, neither the addition of conventional biomarkers in Model 2 to the risk factors in Model 1 nor the subsequent addition of baseline fasting glucose in Model 3 improved the area under the receiver operating characteristics curve (AUC). However, including metabolites in Model 4 increased AUC for IFG and dysglycemia. For IFG, the AUC went from 0.81 in Model 3 to 0.97 in Model 4 (*p* = 0.002). We found a similar pattern for combined dysglycemia. For EFG, inclusion of baseline fasting glucose in Model 3 yielded a marginally higher AUC than Model 2 (difference in AUC: 0.07 [95% CI: 0.00, 0.15]; *p* = 0.06), with an additional marginal increase from Model 3 to Model 4 after including metabolites (difference in AUC: 0.05 [95% CI: −0.01, 0.11; *p* = 0.08]).

In girls, we did not have enough cases of IFG (*n* = 6) or dysglycemia (*n* = 9) to compare across all models, but we observed similar results to that for boys for EFG (Table 3).

Excluding seven participants with IFG at baseline did not change the results (data not shown; available upon request), so we included all participants in the final models.

### 2.3. Results from Project Viva

Table 4 shows the results of the predictive models in Project Viva. Due to the low prevalence of IFG at follow-up (*n* = 1 girl, *n* = 5 boys), we focused on EFG as the outcome. In boys, there was no significant difference in the predictive performance of Model 2 vs. Model 1 or Model 3 vs. Model 2, but, as was the case in EPOCH, including the metabolites in Model 4 significantly increased AUC from 0.66 (Model 3) to 0.84 (Model 4; *p* = 0.003). We observed the same pattern in girls (AUC_Model 4 vs. Model 3_ = 0.89 vs. 0.78; difference in AUC = 0.12 [95% CI: 0.02, 0.22]; *p* = 0.02).

## 3. Discussion

### 3.1. Summary

In the EPOCH youth cohort, we identified sex-specific metabolomic biomarkers that improved the capacity of models to predict impaired fasting glucose and dysglycemia across the 6 years of follow-up, above and beyond known risk factors and biomarkers of T2D risk, including baseline glycemia. When we implemented the predictive models in an independent population of similarly aged youth (Project Viva), we found consistent results with respect to elevated fasting glucose. In boys, the metabolites were part of lipid, amino acid, carbohydrate, nucleotide, and cofactors and vitamin metabolism pathways. In girls, the metabolite predictors were in lipid, amino acid, energy, and nucleotide metabolism pathways.

### 3.2. Biological Relevance of Metabolites

There is growing interest in the use of ‘omics data to identify functional biomarkers of disease risk. Of particular relevance are circulating metabolites that reflect the impact of exposures and experiences that occur “above the skin” to promote disease progression, as well as an individual’s innate physiology “below the skin”, including physiological responses to external stimuli that may be indicative of pathogenic processes [4]. While it is not possible to discern whether metabolites of interest are causally involved in the development of T2D or are simply markers of parallel etiological processes in an observational study, we discuss the biological relevance of metabolites in relation to the pathophysiology of T2D below.

Although most metabolites identified in either sex were on lipid and amino acid metabolism pathways, the specific set of compounds was largely non-overlapping for boys vs. girls. This phenomenon likely reflects sex differences in metabolism. In Project Viva, we have previously reported sex-specific changes in the adipoinsular axis and blood pressure across early adolescence. In comparison to girls, boys exhibited a larger decrement in adiponectin and leptin, two adipose tissue-derived regulators of satiety, weight, and insulin sensitivity; a larger, albeit marginally significant, increase in fasting glucose; and a greater increase in blood pressure from age 7 to 13 years [17]. There is also extensive literature documenting sex differences in metabolism that start as early as in utero via sex-specific metabolic programming [18] and remain apparent throughout the life span, including but not limited to differences in energy metabolism, fat deposition patterns, development of insulin resistance, and metabolic response to adipokines [19]. These nuances of male vs. female metabolism may be drivers or consequences of the sex specificity of the metabolite patterns identified in the present analysis.

However, we noted that three compounds were identified in both sexes: margarate (17:0), orotate, and serine. While we could not find any published papers about margarate (17:0) in relation to metabolic risk, a mechanistic study found that orotate caused endothelial dysfunction in human umbilical vein endothelial cells and induced a 125% increase in insulin resistance in rats [20], and increasing evidence indicates a positive correlation of serine, a non-essential amino acid that plays a central role in a broad range of cellular processes, with insulin secretion and sensitivity [21].

In boys, several metabolites identified in the discovery analysis have been linked to excess adiposity and/or metabolic risk. Noteworthy compounds include the amino acids tyrosine, tryptophan, and leucine, which are part of the BCAA metabolite pattern [8,9,14] that has been implicated in worsening insulin resistance in youth [13] and young adults [7] as well as incident T2D in adults [5,7]. Another important metabolite of interest is glutamate, which surfaced as the strongest metabolite predictor of T2D among a targeted panel of 35 amino acids and acylcarnitines in the Malmö Preventive Project [22].

In girls, the amino acids kynurenate, phenylalanine, and N-acetylvaline have either been identified as part of the BCAA pattern or are derivatives of compounds in this pattern [8,9,14]. Several metabolites were on energy cycling pathways (e.g., malate, succinate, and citrate on the TCA cycle), nucleotide turnover processes (e.g., adenine and beta-alanine involved in nucleotide metabolism), and one-carbon metabolism (e.g., cysteine and N-formylmethionine). Together, these compounds are indicative of rapid cell growth and proliferation that could be linked to the development of chronic disease risk through oxidative stress [23].

### 3.3. Predictive Capacity of Conventional Risk Factors, Biomarkers, and Metabolites

In both EPOCH and Project Viva, the inclusion of conventional biomarkers of T2D risk (baseline waist circumference, fasting insulin and glucose, lipids, Tanner stage, and fasting glucose) to a “base” model comprising several established T2D risk factors (Hispanic ethnicity, family history of T2D, in utero exposure to GDM, baseline BMI) did not improve model performance, nor did the inclusion of baseline glycemia. However, the addition of the metabolomic biomarkers significantly improved the prediction of glycemia outcomes.

These findings dovetail with those from adult studies showing that metabolite profiles predictive of T2D are detectable up to a decade prior to the development of T2D precursors such as elevated fasting insulin and insulin resistance [5,7], and growing literature on children and adolescents showing prospective associations of metabolites with metabolic disease precursors and biomarkers [13,14,16]. Together, current literature suggests that metabolite profiles reflect perturbations to metabolism that occur earlier on the disease progression spectrum and, accordingly, may serve as early flags and/or targets for preventive intervention.

We noted that the metabolites identified herein improved predictive capacity not only for fasting glucose-based outcomes (impaired fasting glucose and elevated fasting glucose) but also for combined dysglycemia—which includes both impaired fasting glucose (IFG) and impaired glucose tolerance (IGT)—among boys in EPOCH. This was somewhat surprising given that different mechanisms may underlie IFG vs. IGT, the latter of which is included in the definition of dysglycemia. IFG is thought to be the result of hepatic insulin resistance, whereas IGT results primarily from reduced peripheral insulin sensitivity [24]. Future studies with more cases of IGT are required to confirm the findings and explore mechanistic differences.

### 3.4. Strengths & Limitations

This study has several strengths. First, we were able to identify metabolomic biomarkers predictive of future impaired fasting glucose and dysglycemia in one cohort (EPOCH) and replicate the results in an independent cohort (Project Viva). Second, both cohorts comprise healthy youth, which is important from the standpoint of preventing the establishment of difficult-to-reverse T2D precursors. Third, we had a relatively large sample size for both cohorts in comparison to other metabolomics analyses in youth (most *N* < 300 [8,25,26,27,28]. Finally, both cohorts are ethnically diverse and located in different geographic regions, which enhances generalizability.

A limitation of this study is the relatively low prevalence of clinically relevant outcomes in EPOCH and Project Viva, which was expected given the relatively healthy status of participants in both cohorts. Additionally, Project Viva did not implement an OGTT, so we were not able to compare model performance for dysglycemia. Regardless, we believe that our findings with respect to elevated fasting glucose are relevant at the population level, given that such differences are indicative of early pathophysiological changes in glucose-insulin homeostasis that have implications for future disease risk. Finally, baseline glucose in Project Viva had higher variability than anticipated, so we have interpreted the results from models with baseline fasting glucose cautiously.

## 4. Methods

### 4.1. Exploring Perinatal Outcomes among Children (EPOCH)

The primary population in this analysis is the Exploring Perinatal Outcomes among Children (EPOCH) study, a prospective cohort of youth in Colorado [29]. At baseline (2006–2009), we enrolled 604 children and adolescents (median age: 10.0 y, range: 6.0–13.4 y) whose mothers were members of the Kaiser Permanente of Colorado Health plan. Of them, 417 mother–child pairs returned for a follow-up visit six years later (median age: 16.3 y; range: 12.6–19.6 y). After excluding children of type 1 diabetic women (*N* = 7), 5 participants without sufficient blood volume at baseline for metabolomics profiling, and 14 with missing data on fasting glucose at follow-up, the analytical sample comprised 391 youth. The Colorado Multiple Institutional Review Board (Protocol #05-0623) approved this study.

All mothers provided written informed consent, and children provided assent or full informed consent if 18 years or older.

#### 4.1.1. T2D Risk Factors at Baseline in EPOCH

At the baseline, participants reported race/ethnicity as non-Hispanic White, non-Hispanic Black, Hispanic, or non-Hispanic other. We collapsed these categories into Hispanic vs. non-Hispanic based on the disproportionately high prevalence of T2D in Hispanic populations and the more frequent occurrence of genetic polymorphisms associated with T2D among persons of Hispanic ethnicity [30].

We calculated BMI using the participants’ weight (kg; measured on a digital scale) and height (cm; measured on a calibrated stadiometer) and standardized values using the World Health Organization (WHO) growth reference [31]. At this time, we also inquired about parental history of T2D.

Information on in utero exposure to gestational diabetes mellitus (GDM) was obtained from the KPCO perinatal records [32].

#### 4.1.2. Conventional Biomarkers of T2D Risk at Baseline in EPOCH

A phlebotomist collected 8-h fasting blood. The blood samples were flash-frozen and stored at −80 °C until time of analysis. We used these biospecimens to assay fasting glucose using an enzymatic procedure and insulin via a radioimmunoassay (Millipore, Darmstadt, Germany). Total cholesterol, HDL, and triglycerides were measured using an Olympus AU400 advanced chemistry analyzer system.

Research assistants measured waist circumference (cm) via a non-stretchable measuring tape [33]. Participants reported their pubertal development based on diagrams of the Tanner stages [34,35]. For the analysis, we focused on pubic hair development, given the relevance of premature pubarche to glucose-insulin homeostasis, especially in females [36].

#### 4.1.3. Untargeted Metabolomics Profiling at Baseline in EPOCH

Metabolon^®^ carried out untargeted metabolomics profiling in fasting plasma collected at baseline using a multi-platform mass spectroscopy (MS)-based technique. The procedure identified 1193 unique features, of which we retained 767 after QA/QC procedures. Details on laboratory procedures and bioinformatics pipelines are published [37].

#### 4.1.4. Glycemia at Follow-Up in EPOCH

At follow-up, we implemented a 2 h 75 g oral glucose tolerance test (OGTT) after a 10 h fast and drew blood at 0, 30, and 120 min post-OGTT for glucose and insulin measurements. We used baseline and 2 h OGTT glucose to derive glycemia outcomes. The primary outcome is impaired fasting glucose (IFG; plasma glucose 5.6–6.9 mmol/L). Second, we assessed elevated fasting glucose (EFG), defined as Q4 vs. Q1 of fasting glucose, to address imbalance issues in the assessment of predictive capacity where model AUC values are inflated because of artificially high specificity arising from the large proportion of non-case participants (i.e., normoglycemic youth). We considered consistency in AUC across both IFG and EFG, in addition to the magnitude of AUC values themselves, for a more robust assessment of predictive capacity, as we have previously done for prior predictive analyses with rare outcomes [37]. Finally, we considered a combined measure of dysglycemia, defined as IFG or impaired glucose tolerance (IGT; glucose 7.8–11.0 mmol/L 2-h post-OGTT) or T2D (fasting glucose ≥ 7 mmol/L or glucose ≥ 11.1 mmol/L at 2-h post-OGTT) [38].

### 4.2. Project Viva

Following the discovery analysis, we sought to assess consistency of findings in an independent sample of youth in Project Viva [39], a pre-birth cohort recruited from a multi-specialty group practice in eastern Massachusetts (Atrius Harvard Vanguard Medical Associates). At the two research visits of interest for this analysis, the participants were median age 12.8 years at the early teen visit (“baseline”; age range 11–16 y) and 17.4 years at the mid-teen visit (“follow-up”; age range 15–20 y) as of March 2021 (research visits are ongoing through Summer 2021).

Of the 2128 mother–child pairs enrolled at birth, 1038 attended the early teen visit. Of the 636 who provided fasting blood at this visit, 560 had adequate fasting plasma volume for untargeted metabolomics profiling [9]. Of them, the present analysis includes 265 youth for whom we currently have fasting blood from the ongoing mid-teen visit and complete data on the conventional risk factors and biomarkers of interest. The Institutional Review Board of Harvard Pilgrim Health Care approved all study protocols. All mothers provided written informed consent, and children provided verbal assent until age 18 years, after which they provided informed consent for themselves.

#### 4.2.1. T2D Risk Factors at Baseline in Project Viva

At the time of enrollment during early pregnancy, we inquired whether the mother herself or the child’s biological father had ever been diagnosed with T2D. Prenatal medical records provided information on perinatal characteristics, including the mother’s prenatal glucose tolerance status [40]. We used this information to classify the children as having a family history of T2D if either the mother or the child’s biological father had a diagnosis of T2D.

At a research visit that took place when participants were 3–6 years of age, we administered a questionnaire to the mothers, inquiring about their child’s race/ethnicity (Black, Hispanic, White, Asian/Pacific Islander, American Indian or Alaskan Native, or Other). For the analysis, we supplemented missing values for the child’s race/ethnicity with the mother’s race/ethnicity. To match what was done in EPOCH, we dichotomized race/ethnicity as Hispanic ethnicity (yes vs. no) if the participant indicated that they were Hispanic.

At the early teen visit (baseline for this analysis), we measured the participants’ weight (kg) via an electronic scale (Tanita Corporation of America, Inc., Arlington Heights, IL, USA) and height (cm) measured using a calibrated stadiometer (Shorr Productions, Olney, MD, USA). We used these values to calculate BMI (kg/m^2^) and age and sex standardized percentiles using the age- and sex-specific World Health Organization (WHO) growth reference for children 5–19 years [31].

#### 4.2.2. Conventional Biomarkers of T2D Risk in Project Viva

At the early teen visit, trained phlebotomists collected an 8 h fasting blood sample from the antecubital vein. All samples were refrigerated immediately, processed within 24 h, and stored at −80 °C until time of analysis. We used the fasting blood to measure plasma glucose enzymatically and to assay insulin using an electrochemiluminescence immunoassay (Roche Diagnostics, Indianapolis, IN, USA). We noted that glucose values in Project Viva participants at baseline (early teen visit) had higher variability than anticipated for fasting values (median: 4.8 mmol/L, range: 3.9–7.6), which may be due to variation in handling of the specimen and/or freeze-thaw cycles prior to glucose measurements. Given that adjustment for baseline fasting glucose was conducted as a sensitivity (as opposed to primary) analysis, we did not exclude any values but, instead, interpreted results cautiously.

Using the same fasting blood sample, we also measured serum total cholesterol, triglycerides, and HDL, which were measured enzymatically with correction for endogenous glycerol, and calculated low-density lipoprotein (LDL) as total cholesterol—HDL—(triglycerides/5).

Research assistants measured the participants’ waist circumference just above the iliac crest to the nearest 1 mm using a Hoechstmass non-stretchable measuring tape (Hoechstmass Balzer GmbH, Sulzbach, Germany). The participants also reported their pubic hair development via a questionnaire with pictorial images of Tanner stages.

#### 4.2.3. Untargeted Metabolomics Profiling in Project Viva

We carried out untargeted metabolomic profiling in fasting plasma via Metabolon^®^’s multi-platform mass spectroscopy-based technique. Details regarding sample preparation and analysis for this population have been published [9,41,42,43]. The laboratory analysis yielded 1135 metabolites, 1005 of which were endogenous compounds. Prior to formal statistical analysis, we imputed missing values for these metabolites as ½ the minimum detected value and log_10_-transformed each compound. We assessed for batch effects via principal components analysis plots but observed no notable clustering by race/ethnicity or sex.

### 4.3. Data Analysis

#### 4.3.1. Step 1: Identify Metabolite Predictors of Fasting Glucose in EPOCH

To identify metabolites of interest, we used rank regression (RRR), as previously described [37]. In random samples of half the N for males and females assessed separately, we entered all 767 metabolites from the untargeted data set as predictors into a model, where continuous natural log-(ln)-transformed fasting glucose at follow-up was the outcome. As is customary, we focused on the first factor as it accounts for the most variance in the outcome [44,45]. Factor 1 accounted for 100% of the variance in ln-fasting glucose, suggesting that the model was overfitted. Thus, we repeated this procedure four more times and focused on metabolites in the top 10% highest regression coefficients in the first factor. We retained compounds meeting this threshold in at least four of the five iterations [37].

#### 4.3.2. Step 2: Associations of Metabolites at Baseline with Fasting Glucose at Follow-Up in EPOCH

We examined associations of each metabolite at baseline with ln-fasting glucose at follow-up using linear regression in the entire sample while accounting for quartiles of the child’s age at baseline as a categorical variable to allow for potential non-linear associations of age with glycemia, difference in age between baseline and follow-up to account for differences tempo of maturation across follow-up, and Hispanic ethnicity.

#### 4.3.3. Step 3: Predictive Capacity of T2D Risk Factors, Conventional Biomarkers, and Metabolites in EPOCH

Finally, we compared the predictive performance of four sequential models using logistic regression. In these models, the outcome was IFG, EFG (Q4 vs. Q1 of fasting glucose), or dysglycemia (IFG, IGT, or T2D). Predictors in Model 1 included known risk factors for T2D: Hispanic ethnicity, family history of T2D, in utero exposure to GDM, and baseline BMI z-score. Model 2 included Model 1 variables plus the following biomarkers of T2D risk measured at baseline: waist circumference, fasting insulin, total cholesterol, HDL, and triglycerides. Model 3 was Model 2 plus baseline ln-fasting glucose. Model 4 was Model 3 plus sex-specific metabolites identified in Step 1. In all models, we accounted for quartiles of baseline age and age difference between baseline and follow-up. The estimate of interest was the difference in AUC for each model vs. the prior model (AUC for Model 2 vs. Model 1, Model 3 vs. Model 2, and Model 4 vs. Model 3) using the Mann–Whitney U-statistic. In sensitivity analyses, we excluded seven participants who had IFG at baseline and assessed for differences in the direction, magnitude, and precision of estimates from the predictive models.

#### 4.3.4. Step 4: Assessment of Models in Project Viva

Using data from Project Viva, we identified metabolites from Step 1 and implemented the analysis in Step 3. The predictive models in Step 3 were similar to those used in EPOCH, with a few exceptions. First, no boys and only two girls had a family history of T2D, so this variable was not included in the predictive models. Second, because Project Viva’s metabolomics data was assayed on a slightly older Metabolon platform than EPOCH, one of the metabolites identified boys in the discovery analysis was not measured in Project Viva (2′-Deoxyuridine); therefore, this metabolite was not included in the predictive models either. Third, due to the low prevalence of IFG (0.8% of girls or *n* = 1, and 3.6% of boys or *n* = 5) and the fact that no OGTT was implemented in this cohort, we focused on EFG as the outcome of interest for the replication analysis.

We carried out all analyses using Statistical Analyses System software (version 9.4; SAS Institute Inc., Cary, NC, USA).

## 5. Conclusions

Metabolomic biomarkers identified herein improved the predictive capacity of T2D precursors beyond usual risk factors and conventional biomarkers. The rising prevalence of early-onset T2D and the known correlations between this metabolic illness and other major chronic conditions on the rise in young people (e.g., non-alcoholic fatty liver disease; cardiovascular disease) should motivate future replication studies.

## Figures and Tables

**Table 1 metabolites-12-00404-t001:** Bivariate associations of background characteristics at baseline (age ~10 y) with fasting glucose at follow-up (age ~16 y) among 391 EPOCH youth.

	*N* ^a^	Mean ± SD Fasting Glucose (mmoL/L) at Follow-Up	*p*-Value ^b^
**Sociodemographic characteristics**			
Sex			0.09
Female	194	5.01 ± 1.70	
Male	197	5.07 ± 0.83	
Age at baseline (years)			0.24
6 to <9 y	67	5.22 ± 2.21	
9 to <10 y	79	4.88 ± 0.52	
10 to <11 y	79	4.93 ± 0.37	
11 to <14 y	165	5.10 ± 1.41	
Race/ethnicity			0.08
Hispanic	139	5.19 ± 1.98	
Non-Hispanic	251	4.95 ± 0.76	
Family history of type 2 diabetes			<0.0001
Yes	60	5.70 ± 3.22	
No	331	4.92 ± 0.41	
**Characteristics at baseline (age ~10 y)**			
Body mass index (BMI) z-score ^c^			0.0002
Underweight (<−2.0)	16	5.02 ± 0.91	
Normal weight (≥−2.0 to ≤1.0)	259	4.91 ± 0.36	
Overweight (>1.0 to ≤2.0)	87	5.07 ± 1.14	
Obese (>2.0)	27	6.18 ± 4.25	
Waist circumference (cm)			0.004
Q1 (median: 54.1)	95	4.89 ± 0.48	
Q2 (median: 59.1)	99	4.95 ± 0.35	
Q3 (median: 65.2)	99	4.87 ± 0.39	
Q4 (median: 78.9)	97	5.44 ± 2.54	
Fasting glucose (mmol/L)			0.0004
Q1 (median: 3.8)	87	4.80 ± 0.37	
Q2 (median: 4.3)	92	4.87 ± 0.47	
Q3 (median: 4.7)	107	5.28 ± 2.24	
Q4 (median: 5.2)	104	5.14 ± 1.03	
Fasting insulin (uU/mL)			0.08
Q1 (median: 4.0)	95	4.93 ± 0.49	
Q2 (median: 7.0)	98	4.92 ± 0.36	
Q3 (median: 11.0)	87	5.04 ± 1.16	
Q4 (median: 18.0)	115	5.23 ± 2.18	
Total cholesterol (mg/dL)			0.92
Q1 (median: 129.0)	95	5.06 ± 1.16	
Q2 (median: 148.5)	96	4.92 ± 0.35	
Q3 (median: 164.0)	100	5.21 ± 2.32	
Q4 (median: 191.5)	100	4.96 ± 0.37	
Triglycerides (mg/dL)			0.65
Q1 (median: 50.0)	94	4.95 ± 0.45	
Q2 (median: 65.0)	96	5.04 ± 1.51	
Q3 (median: 85.0)	102	4.97 ± 0.39	
Q4 (median: 134.0)	99	5.17 ± 2.12	
High density lipoprotein (HDL; mg/dL)			0.07
Q1 (median: 37.0)	93	5.26 ± 2.20	
Q2 (median: 45.5)	94	5.06 ± 1.52	
Q3 (median: 52.0)	100	4.95 ± 0.35	
Q4 (median: 62.0)	102	4.90 ± 0.35	
Tanner stage for pubic hair development			0.48
Stage 1	173	4.95 ± 0.36	
Stage 2	135	5.11 ± 1.83	
Stage 3	58	5.19 ± 1.92	
Stage 4	23	4.95 ± 0.43	

^a^ Totals may not add up to 391 due to missing values. ^b^ From a P-for-linear-trend for ordinal variables; from a Type 3 test for a difference for categorical variables. Fasting glucose is natural-log transformed for use in the regression model that generated the *p*-values. ^c^ According to the World Health Organization (WHO) growth reference for children 5–19 years of age.

**Table 2 metabolites-12-00404-t002:** Identity, superpathway, and subpathway of metabolites assayed from fasting blood at baseline (age ~10 y), selected as predictors of fasting glucose at follow-up (age ~16 y) using reduced rank regression (RRR) in the EPOCH cohort.

Metabolite Name	Superpathway	Subpathway	Average RRR Regression Coefficient
**Boys**			
Leucine	Amino Acid	Leucine, Isoleucine and Valine Metabolism	6.07
Glutamate	Amino Acid	Glutamate Metabolism	4.03
Arginine	Amino Acid	Urea cycle; Arginine and Proline Metabolism	2.96
Tryptophan	Amino Acid	Tryptophan Metabolism	2.32
Margarate (17:0)	Lipid	Long Chain Fatty Acid	2.03
Lactate	Carbohydrate	Glycolysis, Gluconeogenesis, and Pyruvate Metabolism	1.95
N-Acetylvaline	Amino Acid	Leucine, Isoleucine and Valine Metabolism	1.79
Malate	Energy	TCA Cycle	1.59
Caprate (10:0)	Lipid	Fatty acid, Monohydroxy	1.51
Urea	Amino Acid	Urea cycle; Arginine and Proline Metabolism	1.44
Orotate	Nucleotide	Pyrimidine Metabolism, Orotate containing	1.39
Thyroxine	Amino Acid	Tyrosine Metabolism	1.24
N-Formylmethionine	Amino Acid	Methionine, Cysteine, SAM and Taurine Metabolism	1.22
Sarcosine	Amino Acid	Glycine, Serine and Threonine Metabolism	1.05
Quinolinate	Cofactors and Vitamins	Nicotinate and Nicotinamide Metabolism	0.91
Tyrosine	Amino Acid	Tyrosine Metabolism	0.80
2′-Deoxyuridine	Nucleotide	Pyrimidine Metabolism, Uracil containing	0.70
Beta-alanine	Nucleotide	Pyrimidine Metabolism, Uracil containing	0.70
Serine	Lipid	Medium Chain Fatty Acid	0.45
**Girls**			
Glutamine	Amino Acid	Glutamate Metabolism	8.80
Citrate	Energy	TCA Cycle	6.18
N-acetylvaline	Amino Acid	Leucine, Isoleucine and Valine Metabolism	5.36
Myristate (14:0)	Lipid	Long Chain Fatty Acid	5.23
Margarate (17:0)	Lipid	Long Chain Fatty Acid	4.56
Phenylalanine	Amino Acid	Phenylalanine Metabolism	4.19
Kynurenate	Amino Acid	Tryptophan Metabolism	3.61
Chenodeoxycholate	Lipid	Primary Bile Acid Metabolism	3.38
Ornithine	Amino Acid	Urea cycle; Arginine and Proline Metabolism	3.33
Cystine	Amino Acid	Methionine, Cysteine, SAM and Taurine Metabolism	2.82
Serine	Lipid	Medium Chain Fatty Acid	2.58
Adenine	Nucleotide	Purine Metabolism, Adenine containing	1.88
Orotate	Nucleotide	Pyrimidine Metabolism, Orotate containing	1.54
Succinate	Energy	TCA Cycle	0.99

Abbreviations: RRR—reduced rank regression.

**Table 3 metabolites-12-00404-t003:** Comparison of area under the receiver operating characteristic curve (AUC) for conventional risk factors vs. metabolites at baseline (age ~10 y) predicting impaired fasting glucose (IFG), elevated fasting glucose, and dysglycemia at follow-up (age ~16 y) in EPOCH.

	Outcomes at Follow-Up (Age ~16 y)								
	IFG (Yes vs. No)			Elevated Fasting Glucose (Q4 vs. Q1 of Fasting Glucose) ^e^			Dysglycemia (Yes vs. No)		
	AUC	β (95% CI) ^a^	*p*	AUC	β (95% CI)	*p*	AUC	β (95% CI)	*p*
**Boys (*n* = 197)**	***n* = 14 vs. 183**			***n* = 53 vs. 50**			***n* = 18 vs. 179**		
Model 1: Conventional risk factors at ~10 y ^b^	0.65	--	--	0.68	--	--	0.63	--	--
Model 2: Model 1 + biomarkers at ~10 y ^c^	0.74	0.09 (−0.03, 0.21)	0.15	0.73	0.05 (−0.03, 0.12)	0.20	0.68	0.06 (−0.05, 0.17)	0.28
Model 3: Model 2 + fasting glucose at ~10 y	0.81	0.07 (−0.04, 0.19)	0.18	0.80	0.07 (0.00, 0.15)	0.06	0.74	0.06 (−0.07, 0.19)	0.39
Model 4: Model 3 + metabolites at ~10 y ^d^	0.97	0.16 (0.06, 0.27)	0.002	0.86	0.05 (−0.01, 0.11)	0.08	0.89	0.15 (0.03, 0.28)	0.02
**Girls (*n* = 194)**	***n* = 6 vs. 188**			***n* = 54 vs. 56**			***n* = 9 vs. 185**		
Model 1: Conventional risk factors at ~10 y ^b^	--	--	--	0.70	--	--	0.80	--	--
Model 2: Model 1 + biomarkers at ~10 y ^c^	--	--	--	0.72	0.02 (−0.05, 0.08)	0.57	0.93	0.12 (0.00, 0.25)	0.06
Model 3: Model 2 + fasting glucose at ~10 y	--	--	--	0.72	0.00 (−0.01, 0.00)	0.48	0.93	0.01 (−0.02, 0.04)	0.53
Model 4: Model 3 + metabolites at ~10 y ^d^	--	--	--	0.88	0.16 (0.07, 0.26)	0.0007		--	--

^a^ Estimates are a difference in AUC for Model 2 vs. Model 1, Model 3 vs. Model 2, and Model 4 vs. Model 3. ^b^ Includes quartiles of age at baseline; difference in age between baseline and follow-up; ethnicity (Hispanic vs. non-Hispanic), BMI z-score, in utero exposure to gestational diabetes, and family history of type 2 diabetes. ^c^ Model 1 + waist circumference, fasting insulin, total cholesterol, triglycerides, HDL, and Tanner stage for pubic hair development at baseline. ^d^ Model 3 + sex-specific metabolites measured at baseline (shown in Table 3 for boys and Table 4 for girls). ^e^ Boys: median glucose for Q1: 4.6 mmol/L and Q4: 5.4 mmol/L; Girls: Q1: 4.8 mmol/L and Q4: 5.0 mmol/L.

**Table 4 metabolites-12-00404-t004:** Comparison of area under the receiver operating characteristic curve (AUC) for conventional risk factors vs. metabolites at baseline (age ~13 y) predicting elevated fasting glucose (Q4 vs. Q1) at follow-up (age ~18 y) among youth in Project Viva.

	Q4 vs. Q1 of Fasting Glucose at ~18 y ^a^		
	AUC	β (95% CI) ^b^	*p*
**Boys**	***n* = 43 vs. 44**		
Model 1: Conventional risk factors at ~13 y ^c^	0.62	--	--
Model 2: Model 1 + biomarkers at ~13 y ^d^	0.64	0.02 (−0.06, 0.10)	0.64
Model 3: Model 2 + fasting glucose at ~13 y	0.66	0.02 (−0.05, 0.09)	0.51
Model 4: Model 3 + metabolites at ~13 y ^e^	0.84	0.17 (0.06, 0.29)	0.003
**Girls**	***n* = 29 vs. 38**		
Model 1: Conventional risk factors at ~13 y ^c^	0.73	--	--
Model 2: Model 1 + biomarkers at ~13 y ^d^	0.75	0.02 (−0.06, 0.09)	0.59
Model 3: Model 2 + fasting glucose at ~13 y	0.78	0.02 (−0.03, 0.08)	0.37
Model 4: Model 3 + metabolites at ~13 y ^e^	0.89	0.12 (0.02, 0.22)	0.02

^a^ Boys: median glucose for Q1 = 4.9 mmol/L and Q4 = 5.1 mmol/L; Girls: median glucose for Q1: 4.8 mmol/L and Q4: 5.0 mmol/L. ^b^ Estimates are a difference in AUC for Model 2 vs. Model 1, Model 3 vs. Model 2, and Model 4 vs. Model 3. ^c^ Includes quartiles of age at baseline, difference in age between baseline and follow-up, BMI z-score, and in utero exposure to gestational diabetes. ^d^ Model 1 + waist circumference, fasting insulin, total cholesterol, triglycerides, HDL, and Tanner stage for pubic hair development at baseline. ^e^ Model 2 + sex-specific metabolites measured at baseline (shown in Appendix A for boys except for 2′-deoxyuridine and in Appendix A for girls).

## Data Availability

The data presented in this study are available on request from the corresponding author.

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
