# Peer review of "Metabolomic Predictors of Dysglycemia in Two U.S. Youth Cohorts"

_metabolites, 2022, doi:10.3390/metabo12050404_

Round 1

Reviewer 1 Report

The manuscript submitted to Metabolites (metabolites-1700435) titled: "Metabolomic predictors of dysglycemia in two U.S. youth cohorts", presented very important topic related to identify metabolite predictors of dysglycemia in youth. Authors built 4 models based on data from 391 youth. Data are very well presented supported with correct calculations and explanations. Based on results, authors evaluate the extent to which the metabolomic biomarkers improve prediction of dysglycemia beyond known risk factors, conventional biomarkers of T2D risk used in research settings (i.e., waist circumference, fasting insulin, lipid profile, pubertal status), and baseline glycemia (fasting glucose). Additional added value was to assess the consistency of associations in an independent cohort of youth in Massachusetts (Project Viva).  In my opinion project was designed very well and paper is written in correct way, very interesting and presented important findings. 

Author Response

Reviewer #1

  1. Comment: The manuscript submitted to Metabolites (metabolites-1700435) titled: "Metabolomic predictors of dysglycemia in two U.S. youth cohorts", presented very important topic related to identify metabolite predictors of dysglycemia in youth. Authors built 4 models based on data from 391 youth. Data are very well presented supported with correct calculations and explanations. Based on results, authors evaluate the extent to which the metabolomic biomarkers improve prediction of dysglycemia beyond known risk factors, conventional biomarkers of T2D risk used in research settings (i.e., waist circumference, fasting insulin, lipid profile, pubertal status), and baseline glycemia (fasting glucose). Additional added value was to assess the consistency of associations in an independent cohort of youth in Massachusetts (Project Viva).  In my opinion project was designed very well and paper is written in correct way, very interesting and presented important findings. 

Response
: We are grateful for the reviewer’s time, and enthusiastic about the positive comments. We look forward to formally contributing to the literature on metabolomics biomarkers of metabolic conditions in youth upon publication of this work.

Reviewer 2 Report

In this study, Perng et al. performed the EPOCH study to identify the sex-specific metabolomic biomarkers of dysglycemia over 6 years of follow-up among youth in Colorado. The consistency was subsequently evaluated in an independent cohort of youth in Massachusetts in Project Viva. A series of metabolites have been identified, with the corresponding RRR regression coefficient determined. The study was well designed and the manuscript is written clearly. I just have two questions regarding the content of the manuscript:

  1. In Project Viva, elevated fasting glucose (EFG) was used as the outcome as the incidence of IFG was low. Is EFG a widely used criteria? The authors may need to provide relevant references and discussions to justify the use of this outcome. Also, I do not think “Q4 vs Q1 fasting glucose” is clear enough for everyone to understand the criteria of EFG. Some additional description about this term is warranted.
  2. It is quite interesting to find that the composition of the metabolite predictors was largely non-overlapping for boys and girls. The clinical significance of the prediction should be introduced. Also, what might be the cause of this sex-specificity? An in-depth discussion on the underlying causes of this sex-specificity will improve the manuscript.

Author Response

  1. Comment: In this study, Perng et al. performed the EPOCH study to identify the sex-specific metabolomic biomarkers of dysglycemia over 6 years of follow-up among youth in Colorado. The consistency was subsequently evaluated in an independent cohort of youth in Massachusetts in Project Viva. A series of metabolites have been identified, with the corresponding RRR regression coefficient determined. The study was well designed and the manuscript is written clearly. I just have two questions regarding the content of the manuscript.

    Response: We thank the reviewer for their critical appraisal of this work, and respond to their questions/comments below.

  2. Comment: In Project Viva, elevated fasting glucose (EFG) was used as the outcome as the incidence of IFG was low. Is EFG a widely used criteria? The authors may need to provide relevant references and discussions to justify the use of this outcome. Also, I do not think “Q4 vs Q1 fasting glucose” is clear enough for everyone to understand the criteria of EFG. Some additional description about this term is warranted.

    Response: Thank you for the opportunity to clarify. Despite not being a standard clinical threshold, we considered EFG as an outcome in both the discovery (EPOCH) and replication (Project Viva) cohorts for two reasons:
  • Address imbalance issues in the assessment of the predictive models. When outcomes are rare (i.e., <10%), model AUC may become inflated due to artificially high specificity arising from the large proportion of non-cases. By assessing the performance of predictive models for a balanced outcome with equal numbers of cases vs. non-cases (Q4 vs. Q1 of fasting glucose at follow-up), we circumvent this issue. We have previously used this approach for assessing metabolite predictors of non-alcoholic fatty liver disease (NAFLD) by considering a secondary outcome of Q4 vs. Q1 of hepatic fat fraction (Perng et al. JCEM 2020 PMID: 32687159).
  • Low prevalence, and thus small N, of impaired fasting glucose (IFG) in Project Viva (0.8% of girls [n=1] and 3.6% of boys [n=5]) precluded our ability to assess this outcome in regression analyses. Given that no OGTT was implemented in Project Viva, we focused on EFG as the outcome in predictive analyses.

To address the reviewer’s concern, we have included the following text in the Methods section:

Page 9-10 lines 308-314 (description of outcome for discovery analysis): “Second, we assessed elevated fasting glucose (EFG), defined as Q4 vs. Q1 of fasting glucose to address imbalance issues in the assessment of predictive capacity where model AUC values are inflated because of artificially high specificity arising from the large proportion of non-case participants (i.e., normoglycemic youth). We considered consistency in AUC across both IFG and EFG, in addition to the magnitude of AUC values themselves, for a more robust assessment of predictive capacity, as we have previously done for prior predictive analyses with rare outcomes [37].”

Page 12 lines 426-428 (description of outcomes for the replication analysis): Third, due to low prevalence of IFG (0.8% of girls or n=1, and 3.6%, of boys or n=5) and the fact that no OGTT was implemented in this cohort, we focused on EFG as the outcome of interest for the replication analysis.

  1. Comment: It is quite interesting to find that the composition of the metabolite predictors was largely non-overlapping for boys and girls. The clinical significance of the prediction should be introduced. Also, what might be the cause of this sex-specificity? An in-depth discussion on the underlying causes of this sex-specificity will improve the manuscript.

    Response: We agree that this is a point worth elaborating. We have now included the following text in the Discussion:

    Page 7 lines 178-191: Although most metabolites identified in either sex were on lipid and amino acid metabolism pathways, the specific set of compounds was largely non-overlapping for boys vs. girls. This phenomenon likely reflects sex differences in metabolism. In Project Viva, we have previously reported sex-specific changes in the adipoinsular axis and blood pressure across early adolescence. In comparison to girls, boys exhibited a larger decrement in adiponectin and leptin, two adipose tissue derived regulators of satiety, weight, and insulin sensitivity; a larger, albeit marginally significant, increase in fasting glucose; and a greater increase in blood pressure from age 7 to 13 years [16]. There is also a large literature documenting sex differences in metabolism that start as early as in utero via sex-specific metabolic programming [17], and remain apparent throughout the life span, including but not limited to differences in energy metabolism, fat deposition patterns, development of insulin resistance, and metabolic response to adipokines [18]. These nuances of male vs. female metabolism may be drivers or consequences of the sex specificity of the metabolite patterns identified in the present analysis.